# The Anthocyanin Accumulation Related *ZmBZ1*, Facilitates Seedling Salinity Stress Tolerance via ROS Scavenging

**DOI:** 10.3390/ijms232416123

**Published:** 2022-12-17

**Authors:** Jie Wang, Delin Li, Yixuan Peng, Minghao Cai, Zhi Liang, Zhipeng Yuan, Xuemei Du, Jianhua Wang, Patrick S. Schnable, Riliang Gu, Li Li

**Affiliations:** 1Seed Science and Technology Research Center, Beijing Innovation Center for Seed Technology (MOA), Beijing Key Laboratory for Crop Genetic Improvement, College of Agronomy and Biotechnology, China Agricultural University, Beijing 100193, China; 2Department of Agronomy, Iowa State University, 2035 Roy J. Carver Co-Laboratory, Ames, IA 50011-3650, USA

**Keywords:** maize, bronze seed, anthocyanin 3-O-glucosyltransferase, stress tolerance, ROS scavenging

## Abstract

Anthocyanins are a class of antioxidants that scavenge free radicals in cells and play an important role in promoting human health and preventing many diseases. Here, we characterized a maize *Bronze* gene (*BZ1*) from the purple colored W22 introgression line, which encodes an anthocyanin 3-O-glucosyltransferase, a key enzyme in the anthocyanin synthesis pathway. Mutation of *ZmBZ1* showed bronze-colored seeds and reduced anthocyanins in seeds aleurone layer, seedlings coleoptile, and stem of mature plants by comparison with purple colored W22 (WT). Furthermore, we proved that maize BZ1 is an aleurone layer-specific expressed protein and sub-located in cell nucleus. Real-time tracing of the anthocyanins in developing seeds demonstrated that the pigment was visible from 16 DAP (day after pollination) in field condition, and first deposited in the crown part then spread all over the seed. Additionally, it was transferred along with the embryo cell activity during seed germination, from aleurone layer to cotyledon and coleoptile, as confirmed by microscopy and real-time qRT-PCR. Finally, we demonstrated that the ZmBZ1 contributes to stress tolerance, especially salinity. Further study proved that ZmBZ1 participates in reactive oxygen scavenging (ROS) by accumulating anthocyanins, thereby enhancing the tolerance to abiotic stress.

## 1. Introduction

Anthocyanins represent a kind of flavonoid compound as primary pigments of many plants, and they play essential roles in plant defense responses to abiotic stresses, e.g., ultraviolet light, drought, salt, and low temperature, and biotic stresses, e.g., microorganisms and pests [1,2,3,4,5]. Plant anthocyanins have strong antioxidant capacity and are also widely used in health care products and food additives [6]. Plant anthocyanins biosynthesis is mainly regulated by structural genes in its synthetic pathway and regulatory genes affecting its metabolism. The structural genes chalcone synthase (CHS), chalcone isomerase (CHI), flavanone-3-hydroxylase (F3H), flavonoid 3′-hydrolyase (F3′H), dihydroflavonol 4-reductase (DFR), anthocyanidin synthase (ANS), and flavonoid 3-glucosyltransferase (UFGT) can directly encode the enzymes required for anthocyanin biosynthesis. For example, after the introduction of maize *CHS* (*C2*) and *CHI* (*CHI1*) genes into Arabidopsis (*Arabidopsis thaliana*) mutants *tt4* and *tt5*, it was found that the anthocyanin content in both mutants was significantly increased [7]. In addition, impaired function of the *F3H* gene in Arabidopsis reduces its accumulation of flavonoids and anthocyanins [8]. In *Petunia hybrida*, researchers transformed the maize *DFR* gene into the white petunia mutant to produce brick red petunia [9]. The *ANS* gene mutation in *Gentiana scabra* resulted in color-faded phenotype [10]. In grape, alteration of *UFGT* gene expression will lead to the color change of grape pericarp from white to red [11]. Furthermore, transcription factors *MYB*, *bHLH*, and *WD40* can affect anthocyanin synthesis by regulating the expression of structural genes. For example, *IbMYB1* regulates the expression of *CHS*, *CHI*, *F3H*, *DFR*, *ANS*, and *UFGT* to promote anthocyanin accumulation in sweet potato (*Ipomoea batatas*) [12]. In tobacco, the overexpression of *FaMYB1* inhibited expression levels of *ANS* and *UFGT*, resulting in a decrease in anthocyanin content [13]. In maize, the famous color-regulation gene *R1* is a bHLH transcription factors, which is directly bind to the promoter regions of another two color genes *C1* and *BZ1* through its own structure, to activate the expression *C1* and *BZ1*, the binding and activation was completely depend on the interaction between *R1* and *C1* [14]. In *Arabidopsis thaliana*, WD40 family protein *TTG1* regulates the expression level of anthocyanin pathway structural genes by interacting with *TT8* (bHLH family protein) and *TT2* (MYB family protein) [15,16].

Adversity stress environments, including low/high temperature, drought/waterlogging, salinization/acid soil etc., will cause plants to produce excessive ROS in the body during growth and development, and cause different degrees of harm [17,18,19,20]. As a signal molecule in cell defense responses, ROS transmits signals once plants encounter adversity hazards via inducing cellular antioxidant mechanisms to remove excessive ROS, which is toxic to cells [21]. The protective mechanism for effective removal of ROS is classified as enzymatic and non-enzymatic; the former includes superoxide dismutase (SOD), catalase (CAT), peroxidase (POD), and ascorbate peroxidase (APX) etc., typical ROS scavenging enzymes. For example, the overexpression lines of *MnSOD* in *Arabidopsis thaliana* showed stronger salt tolerance, and the enzyme activities of CAT and SOD in plants were significantly enhanced compared with WT [22]. The salt stress tolerance was enhanced in *Atwnk8* mutant, due to the high activities of CAT and POD [23]. The non-enzymatic ROS removal mechanisms include ascorbic acid, glutathione (GSH), carotenoids, tocopherols, and flavonoids etc. These substances directly react with ROS and act as substrates in ROS scavenging [24]. Flavonoids represent one of the secondary metabolites involved in plant development and defense as they have strong antioxidant activity [25].

The accumulation of anthocyanins or flavonoid compounds is directly influenced the by environment. Bhatia et al. reported that the expression of flavonoid synthesis related genes and flavonols content were decreased in *Arabidopsis thaliana HY5* mutants under low temperature, while the capacity of flavonol synthesis was restored after overexpression of *AtHY5* [26]. Meanwhile, the overexpression of *AtMYB12* also contributed to the accumulation of flavonoids under low temperature. After shading treatment of grapes, the transcription level of both anthocyanin synthesis and transport was decreased, together with the reduced anthocyanin content. After the recovery of illumination, the anthocyanin accumulated rapidly [27]. In addition, grape growth in soils with excessive water content was not conducive to the anthocyanin accumulation [28], indicating that the flavonoids are of great significance to plants under adverse environmental conditions.

Here, we identified a maize *bronze* mutant with bronze-colored kernels. We cloned the *Bronze* locus and determined that *Bronze* is allelic to *BZ1*, encodes an anthocyanin 3-O-glucosyltransferase in the nucleus as confirmed via BSR-Seq analysis of gene mapping, allelic test and subcellular localization. We later explored the molecular biology and physiological functions of *ZmBZ1* in seed and seedling development. Finally, we confirmed that the *ZmBZ1* participates in anthocyanins accumulation, which is essential to seed germination and seedling stress tolerance, by activating and increasing the reactive oxygen scavenging (ROS) system.

## 2. Results

### 2.1. A New Isolated Maize Bronze Mutant Deposits Pigment in Organs of Seeds, Anther and Stem

Bronze-colored W22 (designated as original W22, ori-W22) was a typical genetic inbred was widely used for mutator-derived mutant screening. An introgression line (IL) of ori-W22, denoted as modified W22 (mod-W22), showing purple color, was an ideal material for domestication study (Appendix A). To identify genes involved in phenotypic variation and environmental adaptation of W22, we conducted reciprocal-cross of ori-W22 and mod-W22. All the F_1_ progenies for both crossing combinations showed purple seeds, and the self-pollinated F_2_ ears, were segregated with purple and bronze seeds at a ratio of 3:1, indicating that the purple color was attributed to a single dominant gene, while bronze colored mutant account for one quarter of the F_2_ offspring, which resulted from the *Bronze* gene mutation (Figure 1A and Appendix A; Appendix A). The bronze color of mutant separated from an F_2_ ear was observable at 16 DAP in the field condition, and the pigment was firstly deposited in the crown of the seed. Then, the color deepened during the development of the pollinated seeds along with the movement of the milk line (Figure 1B and Appendix A).

To find out how the pigment accumulated, we collected the *bronze* mutant and purple colored wild type (WT) seeds from an F_2_ and separated the pericarp, aleurone layer and endosperm by longitudinally cutting the seeds. Samples were collected from WT and *bronze* mutant at three development stages of 16 DAP, 24 DAP, and 31 DAP. It was found that the pigment formation was occurred mainly at aleurone layer in seeds, there is no pigment observed in pericarp and the embryos (Figure 1C and Appendix A). The pigment first slightly sedimented on top of cutin endosperm of aleurone, where the starchy endosperm shows tighter and crystalline state, and then spread throughout the whole seed along with the disappearance of milk line, which is the sign of seed vigor establishment and seed maturity [29].

To evaluate whether the pigment originated from aleurone layer or from cutin endosperm, the *bronze* and WT seeds at 22 DAP, 26 DAP, and 31 DAP were sliced along the embryo and observed using the stereomicroscope. Clearly, the obvious purple cell layers were appeared in purple seeds from 22 DAP to 31 DAP, while no pigment observed in *bronze* mutants. The layers of aleurone layer cells grow inward and develop into perisperm [30], indicating that the colored endosperm is derived from the inward growth of sub-aleurone layer cells rather than the outer growth of outer-endosperm cells (Figure 1D). These findings are consistent with the results of maize pigment deposition reported by Ming et al. [31]. Besides the seed color, the *bronze* mutant also showed color differences in other tissues, as expressed in stem and anthers. The bronze-colored mutant generates normal plants during the whole developing stage, while purple-colored WTs forms anthocyanin-enriched stem and anthers (Appendix A). Given the apparent difference in pigment accumulation, the total anthocyanins and flavonoids of bronze- and purple-colored seeds with pericarp-removed were determined by chemical method, and the total anthocyanin content in *bronze* mutant was significantly lower than purple-colored WTs, while no significant difference observed in flavonoids content (Figure 1E,F).

### 2.2. Mapping and Cloning of ZmBZ1 in Modified W22

For mapping and cloning of the gene involved in this bronze/purple color in maize, we generated the F_2_ populations with two W22 (ori-W22×mod-W22; or mod-W22×ori-W22) lines. At least 50 pericarp-removed purple and bronze seeds from a segregated F_2_ were pooled at stage of 20 DAP, respectively, total RNA was extracted and sequenced for BSR-Seq analysis. The *bronze* locus was preliminarily mapped into 8.9–17.7 Mb interval of about 8.8 Mb on chromosome 9 (Figure 2A). Within this interval, there are 192 open reading frames (ORFs), and 118 of them are specifically high expressed in seeds, according to the public transcriptional data in maize GDB (www.maizegdb.org/). Meanwhile, the transcriptional level of candidate genes in mapping interval was investigated by comparing the *bronze* mutant and purple WTs. Nine genes were differential expressed including a previous reported *bronze 1* (*bz1*) locus that coincidentally in this interval. The corresponding gene Zm00001d045055 was extremely high expressed in WTs while low expressed in *bronze* mutant, and specifically high expressed in seeds. Sequence alignment revealed that a *Mu* transposon was inserted 214 bp downstream of Zm00001d045055 (*BZ1*) start codon in our *bronze* mutant, of which causes dysfunction of BZ1 (Figure 2C). Transcription level of *ZmBZ1* was analyzed by RT-PCR with the RNA extracted from WTs and *bronze* mutant at 20 DAP. Consistently, *ZmBZ1* was hardly detected in *bronze* mutant while highly expressed in purple WT, which was designated as *Bronze* candidate (Figure 2B).

In order to determine if *ZmBZ1* was the target that causes the non-purple color of *bronze* mutant, two additional *Mu* insertional alleles (*bz1-Mu1* and *bz1-Mu2*) of *ZmBZ1* were ordered from maize GDB. The self-pollination of two heterozygous alleles produced offspring with the segregated bronze and purple-colored seeds of 3:1, and the allelic test of heterozygous *bz1-Mu1/+*, *bz1-Mu2/+* and heterozygous *bronze* (A/a) also showed segregation of 3:1 (purple: bronze), indicating that the *bronze* was allelic to *bz1* (Figure 2F,G). In addition, the genotyping result of the segregated F_2_ with primers flanking the *Mu* insertion of *bz1-Mu1* and *bz1-Mu2*, combined with *Mu* specific primer MuTIR, was consistent with the phenotypic observation, indicated a complete linkage of *Mu* insertions with the *bronze* phenotypes (Figure 2H).

### 2.3. The Nucleus-Located ZmBZ1 (Anthocyanidin 3-O-glucosyltransferase) Is Tissue-Specific Expressed in Seed Aleurone Layer

The maize *ZmBZ1* encodes an anthocyanidin 3-O-glucosyltransferase (UDPG-flavonoid 3-O-glucosyl transferase) that directly related to anthocyanidin synthesis [32]. It contains two exons with an ORF of 1413 bp, encoding 471 amino acids (Figure 3A). To further investigate the evolution of the ZmBZ1 in plants, a phylogenetic tree was constructed by using the full-length protein sequence of ZmBZ1 and its homologous from other organisms. The protein sequences downloaded from public databases (https://blast.ncbi.nlm.nih.gov/Blast.cgi). Homology analysis showed that ZmBZ1 had a very conservative evolutionary pattern in plants, and it possesses high similarity with the homologous in *Arabidopsis thaliana*, *Brachypodium distachyon*, *Triticum aestivum*, *Sorghum bicolor* and *Vitis vinifera*, In *Zea mays*, two paralogs existed, *BZ1* and *BZ1-like* protein, which showed 57.93% sequence similarity (Figure 3B).

To explore the spatial and temporal expression profile of *ZmBZ1*, qPCR was performed in tissues, including root, stem, leaf, bracts, silk, ear, tassel, and seeds. The result showed that *ZmBZ1* was expressed in most of tissues, especially high expressed in seeds (Figure 3C). We further investigated the transcriptional pattern of *ZmBZ1* specific in seeds. qPCR analysis of developing seeds showed that *ZmBZ1* was extremely low expressed in embryo and endosperm, while having a relative high expression level in aleurone layer, and gradually increased along with the seed development (Figure 3D). The above results showed that *ZmBZ1* was specifically expressed in aleurone during seed development, which is consistent with the observation of section of developing seeds (Figure 1C,D).

The ZmBZ1 was possible sub-located in nucleus as predicted by online software package of Cell-PLoc (http://www.csbio.sjtu.edu.cn/bioinf/Cell-PLoc-2/). Transient expression of fusion protein 35S:GFP-ZmBZ1 in maize protoplasts showed that GFP signal was overlapped with red fluorescence and co-localized with nuclear marker Athook, consistent with the co-expression of 35S:YFP-ZmBZ1 and Athook in tobacco leaf epidermal cells (Figure 3E,F). The above results indicated that ZmBZ1 was localized in the nucleus, which is conservative in most monocots and dicots.

### 2.4. Spatio-Temporal Expression Pattern of ZmBZ1 Is Corresponded with Anthocyanin Movement during Seeds Germinating

In order to track if the seed germinability is correlated with pigment transferring, WT and *bronze* seeds were placed on the surface of sandy soil for imbibition and germination at 6, 12, 24, 48, 72, and 96 h respectively. Above samples were cross-sectioned and observing the pigments movement via the stereomicroscope. As our expectation, the purple pigments of WT seeds were concentrated and deposited in thin aleurone layer before imbibition. The aleurone layer moistened and softened during seed imbibition. On one hand, cells nearby the basal endosperm transfer layer (BETL) started absorbing water and then the pigments nearby condensed around the embryo, thereafter gradually diffusing in the radicle with the soften of floury endosperm. On the other hand, the enzymes in seed become active and cell divisions occur rapidly with the seed imbibition. Clear visible pigment movement was observed from aleurone layer to cotyledon of close by, transferred to the coleoptile. The movement of pigments was more obvious with the continuous imbibition (Figure 4A). The transportation of pigments in imbibition seeds during seed germination was exactly synchronous with the auxin movement [33]. However, the transfer of anthocyanin could not be directly observed in the mutant (Figure 4A). In order to investigate the gene transcription profile during seeds imbibition, we extracted RNA from endosperm (include aleurone layer) and embryo from different imbibed stages. qPCR analysis showed that *ZmBZ1* was not apparently different expressed at early stage of 6 h imbibition. After 12 h, *ZmBZ1* showed a rapid up-regulation in WT endosperm, about 4.2 times more than WT embryo, 16.9 and 12.2 times more than endosperm and embryo of *bronze*. With the prolonged imbibition or germination, *ZmBZ1* expression in WT was decreased in endosperm while increased in embryo after the next 12 h. A similar increase was observed in *bronze* embryo, consistent with the rule of pigments movement. After 24 h, *ZmBZ1* was downregulated in both endosperm and embryo of WT and *bronze*, but maintained a relatively high expression level in WT embryo as compared with other tissues (Figure 4B).

### 2.5. ZmBZ1 Is Essential for Stress Tolerance and Was Strongly Induced by Salinity Stress

Previous studies have shown that *Arabidopsis thaliana* UDP glycosyltransferase has a strong response to various environmental stimuli, including low temperature, salinity, drought and ABA stresses [34]. To explore if *ZmBZ1* also plays multi-functions in complex stresses during seedling establishment, we germinated WTs and *bronze* seeds by soaking them with NaCl and PEG solution, respectively, to simulate the salinity and drought stresses in different concentration gradients. After seven days of germination in paper towel, the seedlings showed apparent growth inhibition with the increase of stresses degree, especially when the NaCl concentration was higher than 200 mM and the PEG concentration was higher than 15% (Figure 5A). Root and shoot length were measured to make clear whether the growth of radicle or plantule had been restrained. Compared with non-treatment control, both the shoot and root length of *bronze* seedlings were significantly repressed under treatments of NaCl (50~150 mM) and PEG (15~20%), while the aboveground part (plantule or shoot) was more sensitive to the stress than the underground part (root) (Figure 5B,C). Seed germination was greatly restrained in salinity stress of 250 mM NaCl and drought stress of 15% of PEG in both the WT and bronze mutants.

We also investigated whether the seedling growth was affected in *bronze* mutants or if the *ZmBZ1* can facilitate the stress resistance. Three alleles of *bz1* (*bronze*, *bz1*-*Mu1* and *bz1-Mu2*) were grown in sand and watered with 50 mM, 100 mM, and 150 mM NaCl solution for 14 days, the restrained growth between WTs and three *bz1* alleles was most obvious under 100 mM NaCl treatment (Figure 5D,E). We further recorded the primary root length and lateral roots number under different concentrations of NaCl. Both the above two parameters of three *bz1* mutants were decreased compared with the WTs, which was especially significant under the treatment of 50 mM, 100 Mm, and 150 mM NaCl solution (Figure 5F,G). Total RNA of treated seedlings was extracted and *ZmBZ1* expression was determined by qPCR with non-treatment as control. As shown, *ZmBZ1* was significantly induced in WTs while repressed in three *bz1* mutants under salinity stress, and the expression of *ZmBZ1* was extremely high in WTs under 100 mM NaCl (Figure 5H). The seedling development of both WT and three *bz1* mutants was repressed along with the aggravation of stresses, as reflected by the declined biomass in both WT and three *bz1* mutants compared with the non-treatment control. Meanwhile, three *bz1* mutants were declined more dramatically than the corresponding WTs (Figure 5I). In addition, we also found that anthocyanin in aleurone layer was transferred to radicle and embryo through cotyledon along with the embryogenic cell division during seed germination. The degree of pigments movement in WT seeds was inhibited with the increase of NaCl concentration, while the phenomenon of pigments transportation did not present in mutants (Appendix A).

### 2.6. ZmBZ1 Promoted Seedling Salinity Stress Tolerance by Regulating of Anthocyanin Accumulation

Since *ZmBZ1* was strongly induced under salinity stress and along with the pigment transportation during seed germination, we determined the corresponding product anthocyanin content of *ZmBZ1* with whole germinated seedling (14 days after sowing, 14 DAS) under salinity stress. A significant difference of anthocyanin accumulation between WT and three *bz1* alleles was observed under non- or salinity stress. Consistent with the transcription pattern of *ZmBZ1*, the total anthocyanin content was induced by the salinity stress and with the increase of stress intensity in WTs, showed a highest accumulation at the concentration of 100 mM NaCl, and decreased with growth. Meanwhile, there are no specific rules to follow in the *bronze* mutants (Figure 6A). Genes involving anthocyanin biosynthesis of *F3H*, *DFR,* and *ANS* were selected and analyzed in WT and *bronze* mutants under different salinity stresses. Consistent with the accumulation of anthocyanin, *F3H* showed an up-regulation and then down-regulated in all four samples. A sharp inducement was observed at 100 mM in WTs, while it was slightly induced at 50 mM in three *bronze* alleles. A great discrepancy between WT and mutants were demonstrated at 100 mM NaCl (Figure 6B). Different inducement pattern but similar discrepancy between WT and mutants was observed in the other anthocyanin related genes of *DFR* and *ANS* (Figure 6C,D).

### 2.7. ZmBZ1 Enhanced ROS Scavenging under Salinity Stress

ZmBZ1 (anthocyanin-3-glucosyl transferase) converts cyanidin into any one of several anthocyanins cyanidin-3-glucoside, which has strong antioxidant capacity [35,36]. We determined the reactive oxygen species (ROS) levels of WT and three *bz1* alleles. NBT staining was performed on WT and three *bz1* alleles seedlings at 14 DAS under serial concentrations of salt treatment to reflect the ROS level in cells. As our expectation, the three *bz1* alleles contain more superoxide anions than the WTs, regardless of non-treatment control or different concentrations of NaCl treatment (Figure 7A). Meanwhile, total anthocyanins of each sample were extracted to determine antioxidant activity. As demonstrated, the WT has higher antioxidant capacity than the three *bz1* alleles before and after salt treatment, especially under 100 mM NaCl treatment. This is consistent with the accumulation of anthocyanin content (Figure 7B). The ROS scavenge ability of three *bz1* alleles was lower than the WT with the increase of NaCl concentration, and positively correlated with the growth and development rate of the seedlings.

Three typical ROS scavenging enzymes, catalase (CAT), superoxide dismutase (SOD) and peroxidase (POD), were determined to check if the ROS was removed by the enzymatic pathway besides the anthocyanin associated way. We found that regardless of treatment or non-treatment, the expression of *SOD3* in WT was significantly higher than that in the three *bz1* alleles, and was gradually induced with increasing degree of salinity stress, together with the SOD activity (Figure 7C,D). The *CAT1* and *POX1* showed similar expression patterns with *SOD3* before salt treatment, while being greatly promoted and activated in three *bz1* alleles after salinity stress treatment, which was completely opposite to the expression and enzyme activity pattern of *SOD* (Appendix A–E). DAB staining of hydrogen peroxide activity showed consistent result in regard to the activity of *CAT1* and *POX1* (Appendix A). The above results indicated that there is more than one ROS scavenging mechanism in plants. Moreover, *ZmBZ1* participates in ROS scavenging by means of accumulating anthocyanins, and thus enhances tolerance to abiotic stress. SOD had a synergistic effect on the process of anthocyanin scavenging ROS.

## 3. Discussion

### 3.1. ZmBZ1 Enhanced the Salinity Stress Tolerance in Maize through Atypical ROS Scavenging Way

The *bz1* locus in Maize was identified in the 1930s for its frequent mutation and the typical phenotypic variation of pigments. During the years, studies involving the sequence variation, pigments regulation, and enzyme activities of *bz1* have been published, while no literature has mentioned the gene function in abiotic stress. In this study, we cloned the *ZmBZ1* from W22 and the modified W22 introgression line. Except for the pigments difference, there is almost no difference between two W22 lines in terms of seeds morphology, seedling, and adult-plant development in normal condition. The difference was significant once suffering stresses. It was previously reported that the seedling photosynthetic activity and the development will be inhibited once suffering salinity stress [37]. Overexpression of *SbSAP14* in rice can prevent its leaf tip from yellowing and wilting under salinity stress [38]. Meanwhile, high expression of *MfARL1* alleviated the chlorophyll content and photosynthetic activity of *Arabidopsis thaliana* seedlings under salt stress [39]. However, we did not directly observe that the *bronze* mutant showed the phenotypic decolorization in seedling leaves under salinity stress (Figure 5A,D).

Zhou et al. found that the salinity sensitivity of the *ZmSOS1* mutant was caused by impaired root Na^+^ efflux and increased shoot Na^+^ concentration [40]. Overexpression of *AtNHX5* can efficiently improve the salinity tolerance of soybean by transporting Na^+^ and K^+^ from roots to leaves [41]. In this study, we also found that *ZmBZ1* contributes to the strong salinity tolerance of the purple seeds and corresponding seedlings. In order to testify if the salinity stress tolerance in WT can be attributed to the transported Na^+^ or K^+^ from root to shoot, we measured the K^+^, K^+^/Na^+^ ratio, as well as Na^+^ concentrations in shoots and roots of *bronze*, *bz1-Mu1*, and WT by treated with 100 mM NaCl. Compared with the non-treatment control, the Na^+^ content of shoot and root in all WT and two *bz1* alleles is increased, but the Na^+^ in shoot of WT increase more and faster than that in two *bz1* alleles, and the Na^+^ in WT root decrease faster than that in mutants shoot, indicating that in WT, Na^+^ transfer quickly from the root to shoot and slow down the damage that more Na^+^ does to the root. However, in two *bz1* alleles, the Na^+^ in shoot and root are basically the same, and there is no transport out, as demonstrated, indicating the transport of Na^+^ from root to shoot (Figure 6F). This also explains why the leaves fail to decolorize in two *bz1* mutants.

Previous studies have shown that the equilibrium of Na^+^/K^+^ ratio is crucial for plants salinity stress response, and that salinity-sensitive plants can affect seedling growth due to ion imbalance in stems and roots [42,43]. Taha et al. also confirmed that the high K^+^/Na^+^ ratio in tomato seedlings would contributes to their salinity tolerance [44]. Before salt treatment, the K^+^ content in both WT and two *bz1* alleles shoot is higher than it is in root, and there is no difference between WT and mutants. While after NaCl treatment, the K^+^ of WT was decreased rapidly in shoot with no change in root. The K^+^ of *bz1* mutants shoot remained at a high level, and even higher than that of non-treatment, with a slight increase in root. Therefore, in WT, K^+^/Na^+^ ratio is decreased in shoot while increased in root, due to the transferred Na^+^ from root to shoot, and the reduced K^+^ in shoot. Meanwhile, due to the simultaneously enhanced Na^+^ and K^+^ in seedling, K^+^/Na^+^ ratio is almost unchanged in shoot and root of two *bz1* alleles (Figure 6E,G). The above results indicate that the salinity sensitivity of two *bz1* alleles was caused by reasons other than the imbalance of Na^+^ and K^+^ content.

### 3.2. Anthocyanins Accumulation Was Induced by Extreme Environments and MYB109 and MYBR51

Transcription factors involving in plant abiotic stress resistant have been reported frequently [45,46,47]. Li et al. showed that anthocyanin rhamnosyltransferase UGT79B2 and UGT79B3 were regulated by *CBF1* under cold stress. UGT79B2 and UGT79B3 alleviated their abiotic stress tolerance by regulating anthocyanin accumulation [34]. *RrMYB5* and *RrMYB10* increased anthocyanin accumulation in rose by regulating and activating different flavonoid pathway genes, thus enhancing their tolerance to injury and oxidative stress [48].

R2R3 type MYBs were well studied transcription factors in regulation the flavonoids, lignins, and other secondary metabolites [49]. Here, we screened 12 R2R3-MYBs mentioned in the literatures as possibly binding to the BZ1 and regulating the anthocyanins. Moreover, Yang et al. found that CADR2 (CCAAT-DR1-transcription factor 2) and MYBR51 regulated the *BZ1* by binding at the promoter of *BZ1* as a complex [50]. We tested the binding capacity of 12 R2R3 MYBs and MYBR51 on BZ1 promoter while no binding was observed by testing in Y1H. Some studies in *Arabidopsis thaliana* proved that transcript factor *MYB12/111* was alone or interacted as a complex, binding with DELLA protein and positively regulate multiple target genes of flavonoid biosynthesis [51,52]. Moreover, early studies proved that *C1* (*colored aleurone1*) and *R1 (red1)*, respectively, bound to the promoter region of *BZ1* and regulated its expression, and that the accumulation of BZ1 protein and anthocyanins was regulated by the cold stress [32,53]. Therefore, we speculated that more complicated mechanisms may participate in BZ1 regulation and the consequent stress tolerance. So, we performed binding screen of combinations with above 12 R2R3 MYBs and MYBR51. Finally, we identified an R2R3, MYB109, that could interact with MYBR51 as shown as Y2H and BiFC (Figure 8B,C). We found that MYB109 and MYBR51 separately could not bind to the ZmBZ1, while strong signal was observed once co-transformed with three of them (Figure 8A). The expression level of MYB109 and MYBR51 in WT was higher than that of *bz1* mutant under salinity stress. MYB109 expression in WT was slightly induced under the salinity stress, especially treated with 50 mM NaCl. For MYBR51, the expression was dramatically induced in WT under the salinity stress especially treated with 50 mM and 150 mM of NaCl, and a slight inducement was observed in *bz1* mutant under 50 mM NaCl (Figure 8D), indicating that the complex MYB109/MYBR51 positively regulates the expression of *ZmBZ1*. Further studies, including molecular biology, phenotypic and physiology analysis of how this kind of interaction or binding works, as well as the regulation and performance in double mutants, are required to support our speculation. Meanwhile, Beathard et al. showed that the down-regulation of AtMYB109 resulted in the sensitivity of *Arabidopsis thaliana* seedlings to salinity stress, a crucial piece of information that supported our expectation [54].

Based on our and published results, the following working model was proposed. Once maize seedling was exposed to abiotic stresses, such as salinity and drought, excessive ROS was produced, transcription factor combinations, e.g., MYB109 and MYBR51, inducing and activating the expression of *ZmBZ1* by binding to that promoter region. The normal function of BZ1, accelerating the conversion of cyanidin to cyanidin-3-O-glucoside, which is a kind of antioxidant, plays the role of ROS scavenging in WT, thereby enhancing plant tolerance to salinity and other abiotic stresses (Figure 8E).

## 4. Materials and Methods

### 4.1. Plant Materials

The purple-colored maize W22 that denoted as modified W22 (W22′ or mod-W22) is an introgression line of bronze colored W22 (designated as original W22, ori-W22), was obtained from Professor Xu-Mingliang’s lab at the International Maize Improvement Center in CAU. ori-W22 was ordered from maize COOP stock center including two additional Uniform *Mu* alleles of *bz1-Mu1* (UFMu-10195) and *bz1-Mu2* (UFMu-03854). Mapping population was generated by reciprocal crossing of ori-W22 and mod-W22, followed by self-crossing, the F_2_ generation segregated with 3:1 ratio of purple- and bronze-colored seeds, were used for rough mapping and fine mapping of *bronze* locus. A backcrossed high generation of BC_2_F_2_ with purple colored W22 (mod-W22) was used as a recurrent population. Purple (WT) and bronze colored (*bronze* mutants) seeds segregated from an F_4_ or F_5_ generations were used for physiological experiments including stress treatment. Plant materials were cultivated in Hebei (N′39.485283, E′115.974422) during summer and in Sanya (N′18.247872, E′109.508268) during winter.

### 4.2. Sections and Microscopy Observation

Purple and bronze-colored seeds segregated from an F_2_ were collected from 16 DAP until 31 DAP. The fresh WT and mutant kernels were selected at each stage to observe their pigments accumulation and movement patterns. The samples at 16, 24, and 31 DAP, respectively, were cut longitudinally. The seed coat (pericarp), aleurone layer, and endosperm were separated for further observation. WT and mutant kernels at 22, 26, and 31 DAP were cut into 0.1–0.2 mm slices using hand sectioning, and then observed under a Leica S8 APO stereomicroscope (Leica, Weztlar, Germany) [31,55].

### 4.3. BSR-Seq and Fine Mapping Analysis

Approximately 30 seeds (20 DAP) of purple (WT) and bronze (*bronze* mutant) color were isolated from an F_2_ segregated population. Pericarp removed mutant and WT seeds were pooled together respectively and subsequently ground in liquid nitrogen. Total RNA of two pools was extracted with an RNA extraction kit (Mei5bio, Beijing, China). followed by the QC (quality control) with Agilent2100. Then, we performed pair end sequencing on an Illumina HiSeq4000 platform. Raw sequencing data and clean data with trimmed barcodes and primers were obtained from Annoroad (Beijing, China). Single reads that uniquely mapped to the genome were retained for single nucleotide polymorphism (SNP) calling. Identification of SNPs linked to the target gene was performed following the parameters described by published method [56].

### 4.4. RNA Extraction and qRT-PCR Analysis

Total RNA was extracted from various tissues as described above. First-strand cDNA was synthesized using a StarScript Ⅱ RT Mix with gDNA Remover kit (GenStar, Beijing, China). qRT-PCR was carried out in triplicate for each sample using 2 × HQ SYBR qPCR Mix (Zomanbio, Beijing, China) on an ABI Life Q6 real-time fluorescent quantitative PCR instrument (Applied Biosystems, Waltham, MA, USA). The *GAPDH* was used as an internal control. The primer sequences used are listed in Appendix A.

### 4.5. Subcellular Localization

The open reading fragment (ORF) of *ZmBZ1* was amplified from mod-W22 and cloned into 16318hGFP and pEarlyGate 101 vectors (Invitrogen, Carlsbad, CA, USA; Vazyme, Beijing, China). The 35Spro:BZ1-YFP construct was expressed in maize protoplasts and incubated at 25 °C in the dark for 12–16 h. 35Spro:BZ1-YFP was subsequently transformed into Agrobacterium strain GV3101 and expressed in tobacco leaves. The fluorescence signals were analyzed under a Zeiss 880 confocal microscope after 48 h of incubation (Jena, Germany). The constructed 35Spro:BZ1-16318hGFP fusion protein was also transformed into maize leaf protoplasts together with the nuclear protein marker Athook, and the results were observed under the confocal microscope (Zeiss LSM880) 13 h after transformation. Tobacco was grown at 25 °C in 16/8 h of light/dark (white fluorescent light, 10,000 lux) [57].

### 4.6. Phylogenetic Analysis

The predicted amino acid sequences of ZmBZ1 and its homologs in *Zea mays*, *Sorghum bicolor*, *Oryza sativa*, *Arabidopsis thaliana*, *Mus Pahari*, and *Homo sapiens* were downloaded from the BLAST tool of the National Center for Biotechnology Information. An unrooted maximum likelihood (ML) tree was constructed with the software MEGA 6.0 (https://www.megasoftware.net/) using the best model LG with a bootstrap test (1000 replicates).

### 4.7. Assays for Stress Treatment

The NaCl and PEG treatments were conducted according to published methods [58,59]. Paper towels were pre-wetted in water, NaCl, or PEG6000 solutions at different concentrations (50 mM, 100 mM, 150 mM, 200 mM, 250 mM NaCl or PEG6000 of 10%, 15%, and 20%). Hence, 300 seeds of 3 replicates were used for germination with two layers of pre-wetted paper towels under condition of 25 °C, 16 h-light/8 h-dark, non-treatment with double distilled water was used as control. The shoots and roots were measured after 7 days cultivation. For soil medium culturing, sandy soil was mixed with NaCl solutions (50 mM, 100 mM and 150 mM) directly in a ratio of 6.25:1 (v/m). Seeds sterilized with 1% sodium hypochlorite were sown in the prepared medium and cultured under the same conditions as the above paper method. Hence, 100 mL of corresponded NaCl solution was added to the tray of the experimental basin every 4 days. Recording root length, root number and biomass at 14 DAS and the corresponding total RNA of mutant and WT seedlings were extracted and gene expression determined via qRT-PCR analysis.

### 4.8. Quantification of Na^+^ and K^+^ Concentration

Sandy cultured seedlings (14 DAS) were collected for Na^+^ and K^+^ quantification before and after salt treatment. the shoots and roots were dried at 110 °C for 2 h and then at 80 °C for 48 h to constant weight. The dry matter was treated with a muffle furnace at 575 °C for 6 h, and then dissolved the powder in 0.1 N HCl solution. The concentrations of Na^+^ and K^+^ were measured on an atomic absorption spectrometer TRACE AI1200 (Aurora Group Company, Vancouver, BC, Canada) [60,61].

### 4.9. Quantification of Anthocyanins

To observe the transfer of anthocyanin during seeds germinating, the WT and *bronze* mutant kernels were placed on the surface of wetting sandy for imbibition, and samples were collected at 6 h, 12 h, 24 h, 48 h, 72 h, and 96 h during germination. The seeds were cut longitudinally and observed with a stereo microscope. Meanwhile, tissues of embryo and endosperm (aleurone layer) at above germination time points were collected for RNA extraction and *ZmBZ1* expression analysis. Salt treated 14-day seedlings were collected for total anthocyanin extraction and quantification on an infinite M200 Pro microplate reader (Tecan, Männedorf, Switzerland) according to a published method [62].

### 4.10. NBT and DAB Staining

NaCl treated seedling leaves (14 DAS) were stained with nitrobluetetrazolium (NBT) to observe the superoxide as described by Xiong et al. [63]. The same samples were stained with 3, 3-diaminobenzidine (DAB) for measuring the hydrogen peroxide content.

### 4.11. Determination of Antioxidant Capacity

Total anthocyanin was extracted from 14-day seedlings. The total antioxidant capacity of different samples was determined according to instruction of FRAP kit (Beyotime, Shanghai, China) [64]. Fresh seedlings of 0.5–1 g were cut into pieces and placed in a pre-cooled mortar, then thoroughly ground with 15 mL 50 mM PBS (pH 7.0) buffer on ice. The homogenate was centrifuged at 4000× *g* for 10 min. The supernatant was used for enzyme activity determination, including SOD, CAT, and POD, according to the improved method of Jung [65].

### 4.12. Y1H, Y2H and BiFC Assays

The 1 kb upstream sequence from the start codon of *ZmBZ1* was amplified and cloned into pAbAi vector, followed by the linearization of the constructed pBZ1-AbAi plastid digested with restriction enzyme BstBI. Then, the sequence was transformed into Y1H Gold competent cells and cultured on SD/-Ura medium at 30 °C for 2–4 days. Y1HGold[pBZ1-AbAi] competent cells were prepared according to instruction of Frozen-EZ Yeast Transformation II Kit (Zymo Research, Los Angeles, CA, USA). The ORFs of MYB109 and MYBR51 were cloned into pGADT7 vector followed by transformation into yeast strain Y1HGold[pBZ1-AbAi], and cultured on SD/-Leu medium, selecting positive clones and suspending with 0.9% NaCl solution, and cultured on SD-Leu with aureobasidin A (AbA) medium at 30 °C for 3 days. The reporter gene was activated by cell growth test [66].

For the yeast two-hybrid experiment, the ORFs of MYB109 and MYBR51 were cloned into pGBKT7 and pGADT7 vectors, respectively, followed by co-transformation into yeast strain MaV203. Cells were grown on SD/-Leu-Trp selective medium, and then SD/-Leu-Trp-His-Ade medium which contain 3-amino-1,2,4-triazole (3-AT), at 30 °C for 2–4 days.

For bimolecular fluorescence complementation experiment (BiFC), the ORFs of MYB109 and MYBR51 were inserted into pEarleygate201 and pEarleygate202 vectors containing nYFP and cYFP, respectively, and then transformed into Agrobacterium GV3101. Following injection into tobacco leaves for 48 h, the co-expressed YFP (yellow fluorescent protein) fluorescence signal was detected on a Zeiss LSM880 fluorescence microscope. The primers used for Y1H, Y2H, and BiFC detection are shown in Appendix A.

## Figures and Tables

**Figure 1 ijms-23-16123-f001:**
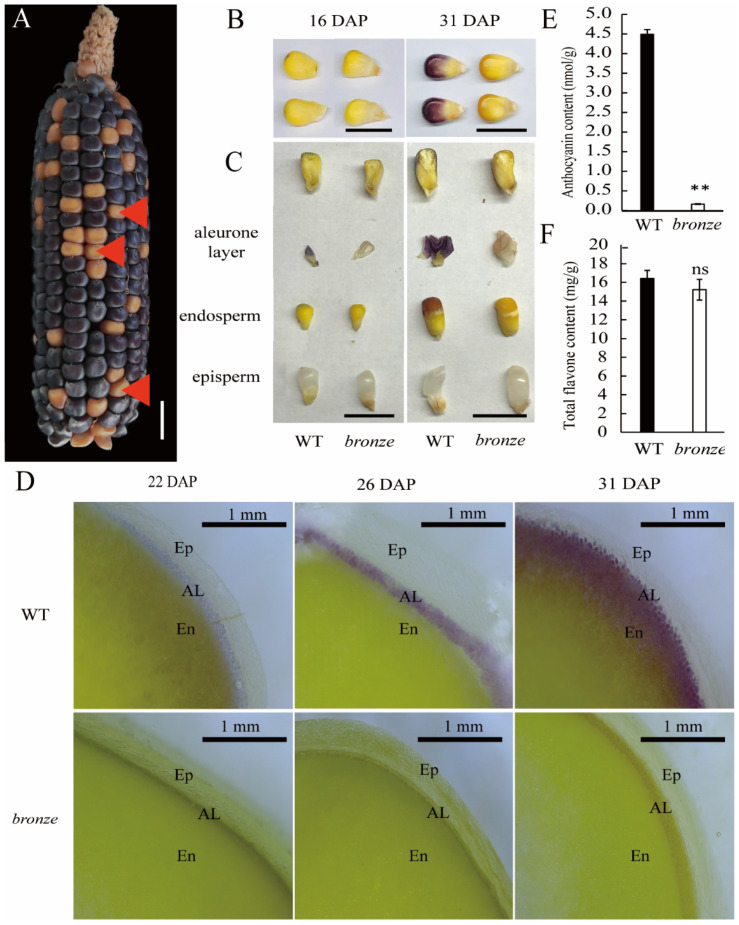
Phenotypic analysis of WT and *bronze* mutant. (**A**) Mature ear of F_2_ from selfed F_1_ (*bronze*/+) plant. The mutants (bronze colored seed) were labeled as red solid triangles. Bar = 1 cm; (**B**) Developing seeds of WT (purple colored) and mutants at 16 and 31 DAP. Bar = 1 cm; (**C**) Separated parts of above developing seeds by longitudinal sections, including whole seeds, separated aleurone layer, endosperm and the seed coat (pericarp) for WT and mutant at the stage of 16 and 31 DAP. Bar = 1 cm; (**D**) Slicing of developing seeds at 22, 26 and 31 DAP photographing on a stereomicroscope. Ep, Episperm; Al, aleurone layer; En, endosperm. Bar = 1 mm; (**E**,**F**) Anthocyanins content (**E**) and total flavonoids content (**F**) of WT and mutant seeds. The asterisk indicates a significant difference relative to WT (Student *t*-test: ** *p* < 0.01; ns means no significant difference).

**Figure 2 ijms-23-16123-f002:**
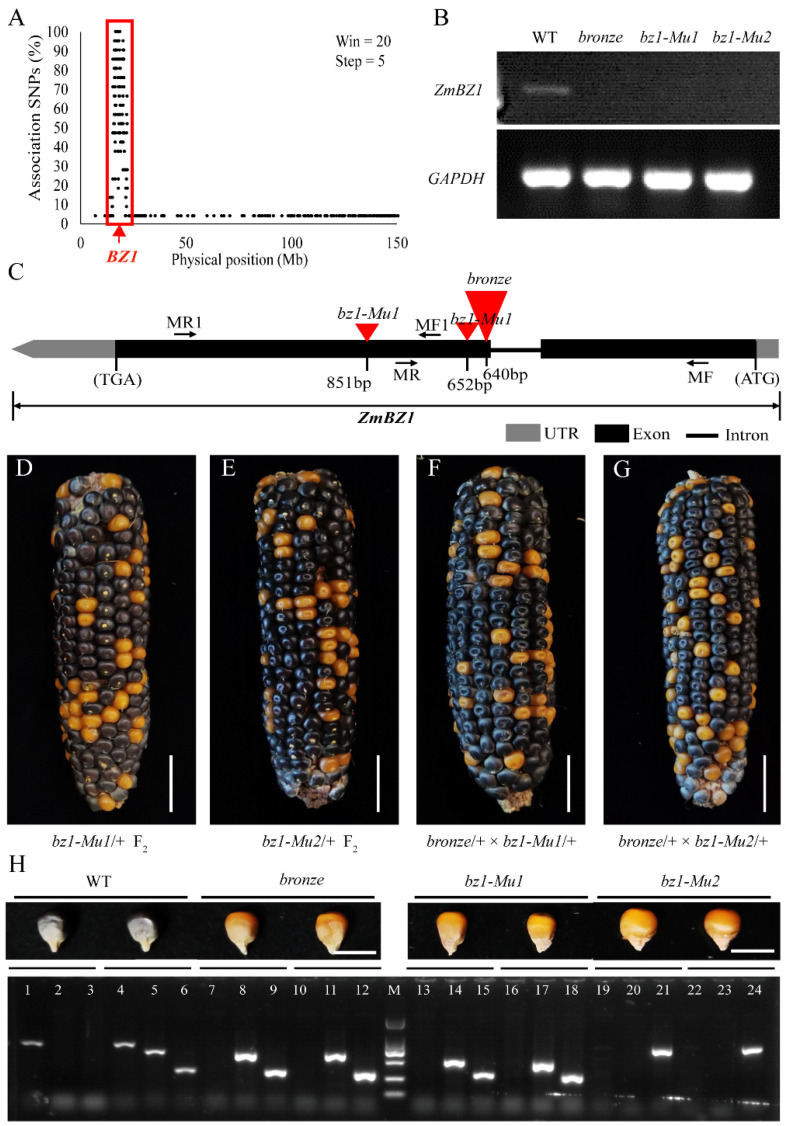
Positional cloning of *ZmBZ1*. (**A**) Raw mapping with BSR-Seq analysis. The *ZmBZ1* locus was mapped to an 8.8 Mb (from 8.9 Mb to 17.7 Mb, Maize AGPv3) interval on chromosome 9; (**B**) RT-PCR analysis of developing seeds (30 DAP) from WT, *bronze*, *bz1-Mu1* and *bz1-Mu2*. *GAPDH* was used as an internal control; (**C**) Gene structure of target gene, the three independent *Mu* insertional alleles that labeled as triangles and the primers used for genotyping were denoted as arrows; (**D**,**E**) Two additional alleles (*bz1-Mu1* and *bz1-Mu2*) of *bronze* mutant (demonstrated with the segregated F_2_ ears); (**F**,**G**) Allelic test of the two additional alleles by crossing the heterozygous *bronze* mutant to each of the heterozygous alleles (*bz1-Mu1*/+ and *bz1-Mu2*/+, respectively). Bar = 1 cm; (**H**) Phenotypic and genotypic analysis of *bronze*, *bz1-Mu1* and *bz1-Mu2* mutants by comparison of WTs. Bar = 1 cm. Primer pairs of MF/MR, MF/MuTIR and MR/MuTIR were used for genotyping of *bronze* and *bz1-Mu1* alleles, primer pairs of MF1/MR1, MF1/MuTIR and MR1/MuTIR were used for genotyping of *bz1-Mu2* allele. Homozygous WT (+/+) was demonstrated in channels #1–3; heterozygous WT (*Mu*/+) was demonstrated in channels #4–6; homozygous mutants (*Mu*/*Mu*) were demonstrated as channels #7–24.

**Figure 3 ijms-23-16123-f003:**
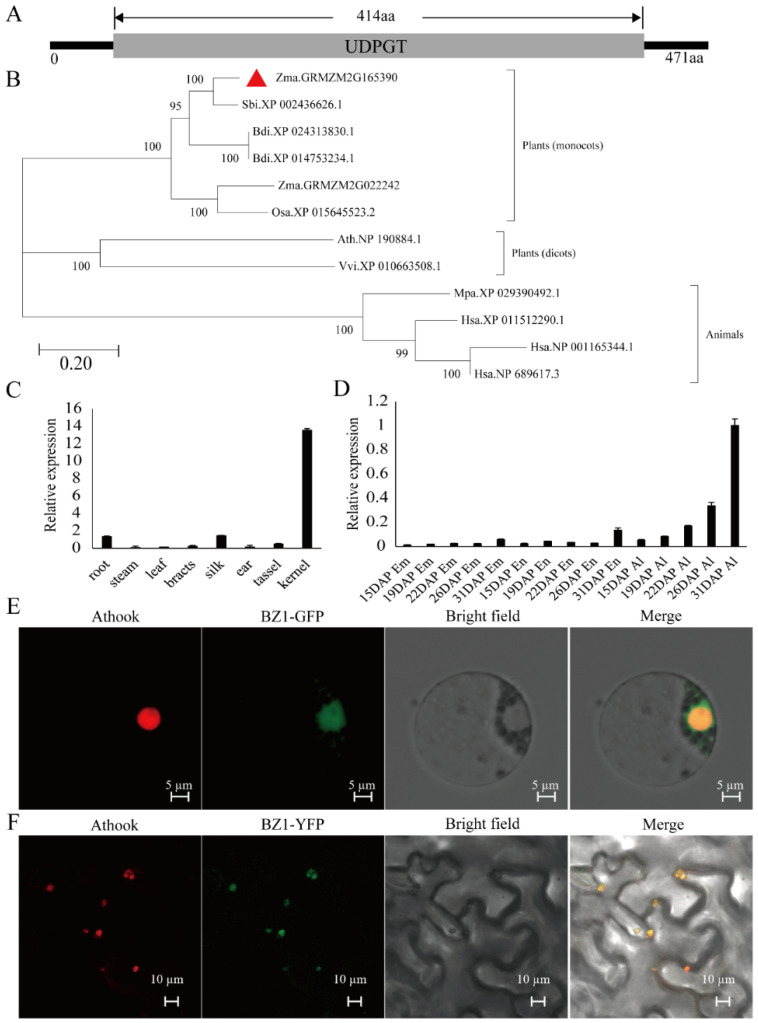
Phylogenetic analysis, spatial and temporal expression profile and subcellular localization of ZmBZ1. (**A**) Protein structure of ZmBZ1; (**B**) Phylogenetic analysis of ZmBZ1 and its homologs in different species. Red triangle protruding ZmBZ1. Bar = 0.2 substitutions per amino acid position; (**C**,**D**) Spatial and temporal expression profile of *ZmBZ1* analyzed via qPCR in various tissues (**C**) and during seed development (**D**). *GAPDH* was used as an internal control. Values are shown as means ± SE, *n* = 3; (**E**,**F**) Subcellular localization of ZmBZ1 protein in maize leaf protoplasts (**E**) and tobacco leaf epidermal cells (**F**). GFP, green fluorescence protein; YFP, yellow fluorescence protein; Athook, a nuclear marker. Bar = 5 μm and 10 μm.

**Figure 4 ijms-23-16123-f004:**
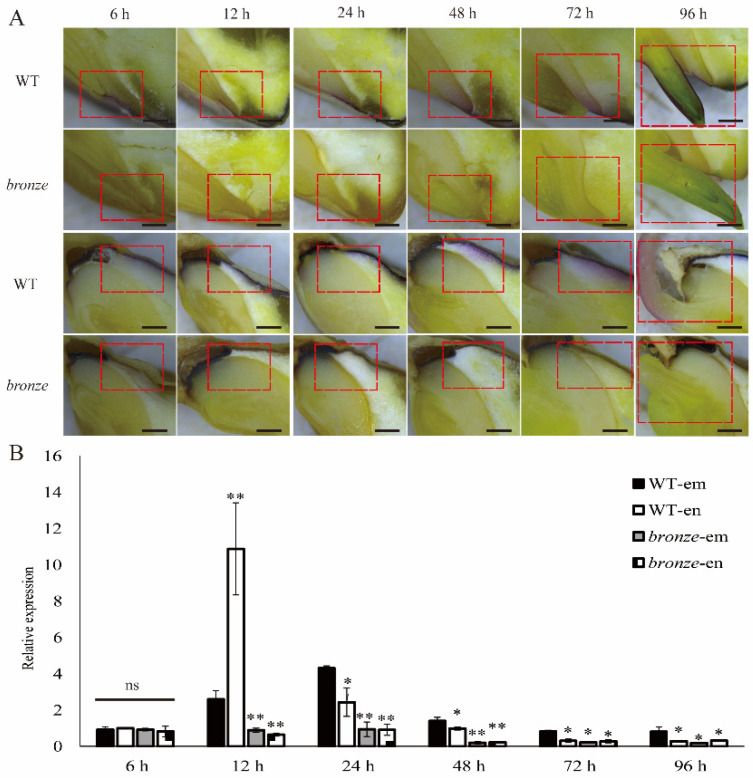
Anthocyanin movement and *ZmBZ1* expression pattern during seed germination. (**A**) Observation of the anthocyanins movement from longitudinal sections of WT and *bronze* seeds that germinated for 6 h, 12 h, 24 h, 48 h, 72 h and 96 h. Red dashed rectangles highlight the location of anthocyanin accumulation. Bar = 1 mm; (**B**) qPCR analysis of *ZmBZ1* expression of WT and *bronze* seeds during germination. *GAPDH* was used as an internal control. Values are shown as means ± SE, *n* = 3. Student *t*-test: * *p* < 0.05; ** *p* < 0.01; ns means no significant difference.

**Figure 5 ijms-23-16123-f005:**
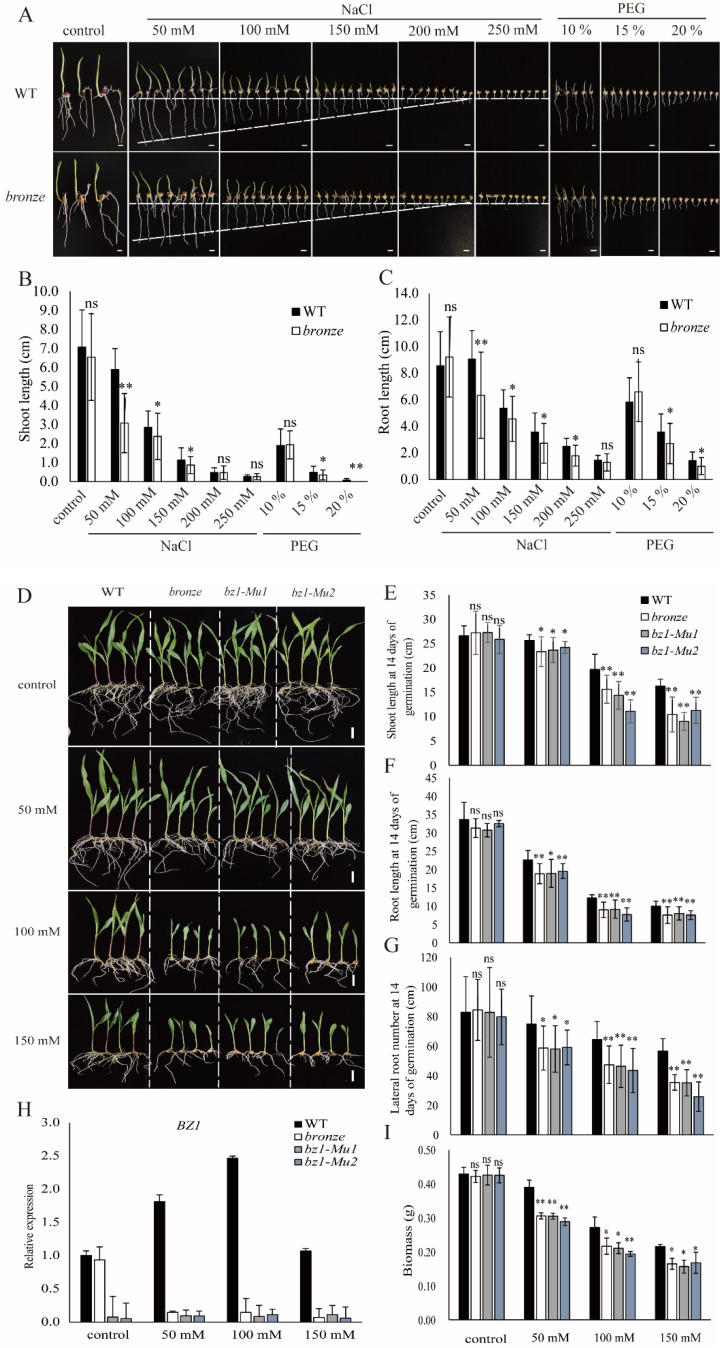
Roles of ZmBZ1 in abiotic stresses tolerance. (**A**–**C**) Simulation of salinity and drought stresses with 50 mM, 100 mM, 150 mM, 200 mM and 250 mM NaCl solutions, and 10 %, 15 %, 20 % PEG solutions, respectively (**A**). The comparison of WT and *bronze* to abiotic stresses was evaluated by measuring the shoot length (**B**) and root length (**C**) of the seedlings germinated for 7 days. Asterisks indicate significant differences relative to the WT (Student’s *t*-test: ** p <* 0.05; *** p <* 0.01; ns means no significant difference); (**D**–**I**) Salinity stress treatment of WT and three *Bz1* alleles by treated with 50 mM, 100 mM and 150 mM NaCl solution for 14 DAS in sands (**D**), shoot length (**E**), primary root length (**F**), number of lateral roots (**G**) and biomass (**I**) of the above seedlings. qPCR analysis of *ZmBZ1* expression in different degree of salinity stress (**H**). Asterisks indicate significant differences relative to non-treatment (Student’s *t*-test: * *p* < 0.05; ** *p* < 0.01; ns means no significant difference).

**Figure 6 ijms-23-16123-f006:**
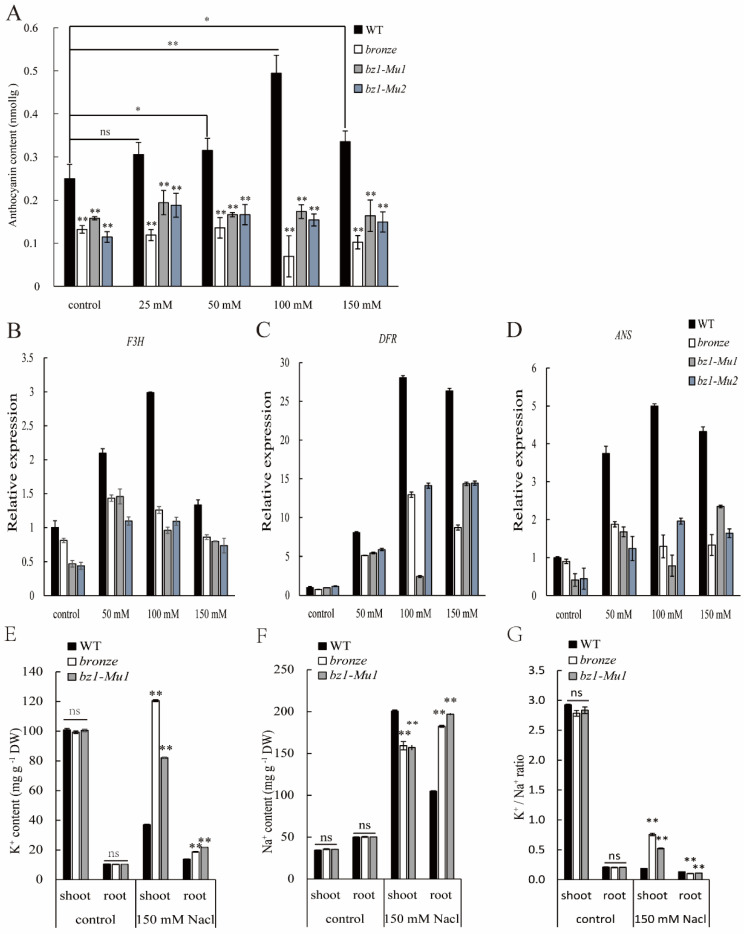
Anthocyanin accumulation was affected by altered *ZmBZ1* expression. (**A**) The content of anthocyanins in three *bz1* alleles and WT seedlings before and after salinity stress treatment. The asterisks indicate the significant difference relative to WT (Student t-test: ** p <* 0.05; *** p <* 0.01; ns means no significant difference); (**B**–**D**) qPCR analysis of *F3H* (**B**), *DFR* (**C**) and *ANS* (**D**) expression in three *bz1* alleles and WT seedlings before and after salinity stress treatment. *GAPDH* was used as an internal control. Values are shown as means ± SE, n = 3. (**E**,**G**) The content of K^+^ (**E**), Na^+^ (**F**) and K^+^/Na^+^ (**G**) ratio in WT, *bronze* and *bz1-Mu1* seedlings before and after salinity stress treatment. The asterisks indicate the significant difference relative to WT (Student *t*-test: * *p* < 0.05; ** *p* < 0.01; ns means no significant difference).

**Figure 7 ijms-23-16123-f007:**
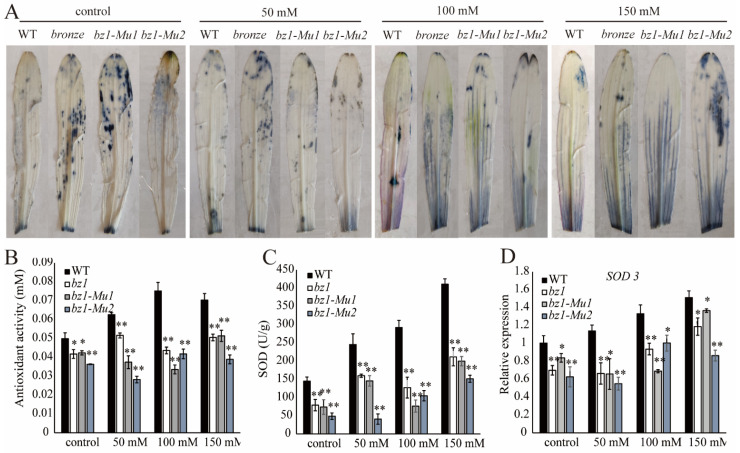
Antioxidant activity conferred by ZmBZ1 and enhanced under salinity stress. NBT staining (**A**), Antioxidant activity (**B**), SOD activity (**C**) and *SOD3* expression (**D**) of three *bz1* alleles and WT seedlings at 14 DAS before and after NaCl treatment. *GAPDH* was used as an internal control. Values are shown as means ± SE, n = 3. Asterisks indicate significant differences relative to the WT (Student’s *t*-test: ** p <* 0.05; *** p <* 0.01).

**Figure 8 ijms-23-16123-f008:**
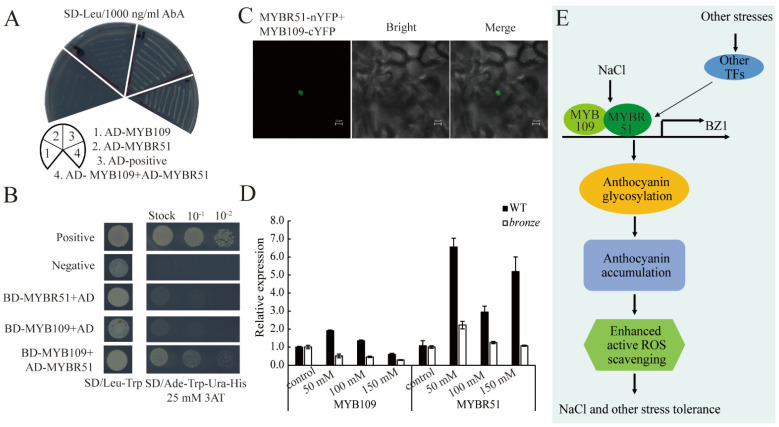
ZmBZ1 was regulated by complex of MYBR51/MYB109. (**A**) Y1H showing that MYB109 and MYBR51 work together on the *ZmBZ1* promoter; (**B**,**C**) Transcription factors MYBR51 and MYB109 were physically interacted with each other by conformation with Y2H and BiFC. Bar = 10 μm; (**D**) *MYB109* expression in WT was slightly induced by the salinity stress, especially under 50 mM NaCl; For *MYBR51*, the expression was dramatically induced in WT by the salinity stress especially under 50 mM and 150 mM of NaCl, a slight inducement was observed in *bronze* mutant under 50 mM NaCl; (**E**) Schematic model of the genetic and environmental regulation of the anthocyanins in stress tolerance.

## Data Availability

All of the data generated or analyzed during this study are included in the published article.

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
