# Peer review of "The Anthocyanin Accumulation Related ZmBZ1, Facilitates Seedling Salinity Stress Tolerance via ROS Scavenging"

_ijms, 2022, doi:10.3390/ijms232416123_

Round 1
Reviewer 1 Report
Introduction has enough information about, the anthocyanin accumulation related ZmBZ1, facilitates seedling salinity-stress tolerance via ROS scavenging, and Methodology for the present study is well articulated. The author has written the results with novel information, contribute to the advancement of understanding and are pragmatic. Discussion of the manuscript is well written. In overall manuscript I will suggest that please include
1. Specific, detailed comments regarding the originality, scientific quality.
2. Check the needs for tables and figures and the adequacy of the references.
3. Check the spellings.
The present work is novel and well written.
Author Response
Dear reviewer, Thank you very much for your highly remark of our work. Below are the responses according to above suggestions: 1. It’s well known that anthocyanins are kind of antioxidant. And studies involving in anthocyanins-dependent stress tolerance have been reported a lot, but few explained the molecular mechanisms. For the “Specific, detailed comments regarding the originality, scientific quality”. We addressed in the discussion part, of which we described that “ZmBZ1 enhanced the salinity stress tolerance in maize through atypical ROS scavenging way”, which means that traditionally, plants salinity stress tolerance is the typical K+/Na+ flux mechanism, while the anthocyanin mediated ROS-scavenging in stress tolerance, especially in salt stress tolerance, it was the originality of our study. Besides, we connected the anthocyanins with the seed germination by real-time tracing of the anthocyanins during seed development and germination. This is also our originality. In order to avoid the article wording too absolutist, we do not use the words too specific, please understand. But we corrected the other mistakes/errors (including spelling, typo) strictly according to reviewer’s suggestion. Please see the revised manuscript (red highlighted).Reviewer 2 Report
This study characterized a maize Bronze gene (BZ1), which encodes an anthocyanin 3-O-glucosyltransferase. The manuscript demonstrated that the ZmBZ1 is essential to seedling stress tolerance, by activates and increase the reactive oxygen scavenging (ROS) system. The results is meaningful for our understanding of the relationship between thocyanins accumulation and aboitic stress in maize.
Some minor comments:
Check the data analysis results. For example, Fig5B, NaCl 100 and 150 mM, are there significant differences between the WT and mutunt? I suggest authours use one-way ANOVA with an LSD test, but not the Student’s t-test in statistical analyses.
Check the Reference format, such as 9, 24, 32, and so on.
Author Response
Dear reviewer, Thank you very much for your hard work and the constructive suggestion. Below are the responses according to above suggestions: 1. For the data analysis, we tried both statistic methods of Student’s test, as well as one-way ANOVA with an LSD test. Both the methods give exactly same results of significance. Take the Figure 5B as example, we re-analyzed the data and please see the attached original data and the each steps of analysis (excel file: Name: Figure 5B-data analysis). Since it’s a huge work to replacing all the figures with a different way but show same result, we just show you attached case of Figure 5B, please let me know if this is acceptable, otherwise, we would like to replace all the figures with one-way ANOVA. But thank you very much for the great suggestion, we will submit other work/manuscripts with updated one-way ANOVA statistics. 2. For the references, Thanks a lot for the reminding, we corrected all the capital characters into the normal lowercase (please see 9, 24, 32, 53, 59, 60, 63 that highlighted with red color). Please see the attachment excel for case of data analysis results in Figure 5B.